**Investigation**

# Neuron-specific repression of alternative splicing by the conserved CELF protein UNC-75 in *Caenorhabditis elegans*

Pallavi Pilaka-Akella,[1] Nour H. Sadek,[1] Daniel Fusca (ID),[2] Asher D. Cutter (ID),[2] John A. Calarco[1,*]

[1]Department of Cell and Systems Biology, University of Toronto, 25 Harbord Street, Toronto, Ontario, Canada M5S 3G5
[2]Department of Ecology and Evolutionary Biology, University of Toronto, 25 Wilcocks Street, Toronto, Ontario, Canada M5S 3B2

*Corresponding author. Department of Cell and Systems Biology, University of Toronto, 25 Harbord Street, Toronto, Ontario, Canada M5S 3G5. Email: john.calarco@utoronto.ca

Tissue-regulated alternative exons are dictated by the interplay between *cis*-elements and *trans*-regulatory factors such as RNA-binding proteins (RBPs). Despite extensive research on splicing regulation, the full repertoire of these *cis* and *trans* features and their evolutionary dynamics across species are yet to be fully characterized. Members of the CUG-binding protein and ETR-like family (CELF) of RBPs are known to play a key role in the regulation of tissue-biased splicing patterns, and when mutated, these proteins have been implicated in a number of neurological and muscular disorders. In this study, we sought to characterize specific mechanisms that drive tissue-specific splicing in vivo of a model switch-like exon regulated by the neuronal-enriched CELF ortholog in *Caenorhabditis elegans*, UNC-75. Using sequence alignments, we identified deeply conserved intronic UNC-75 binding motifs overlapping the 5′ splice site and upstream of the 3′ splice site, flanking a strongly neural-repressed alternative exon in the Zonula Occludens gene *zoo-1*. We confirmed that loss of UNC-75 or mutations in either of these *cis*-elements lead to substantial de-repression of the alternative exon in neurons. Moreover, mis-expression of UNC-75 in muscle cells is sufficient to induce the neuron-like robust skipping of this alternative exon. Lastly, we demonstrate that overlapping an UNC-75 motif within a heterologous 5′ splice site leads to increased skipping of the adjacent alternative exon in an unrelated splicing event. Together, we have demonstrated that a specific configuration and combination of *cis* elements bound by this important family of RBPs can achieve robust splicing outcomes in vivo.

Keywords: alternative splicing; RNA binding proteins; RNA processing; *C. elegans*; gene expression; UNC-75; CELF proteins; WormBase

## Introduction

Alternative splicing (AS) is a co-transcriptional mechanism that drives proteomic diversity, cellular complexity, animal development, and tissue-specificity by generating multiple isoforms from a single precursor mRNA (Yeo *et al.* 2004; Keren *et al.* 2010; Nilsen and Graveley 2010; Mazin *et al.* 2021). It is estimated that 95% of human multiexon protein-coding genes contain at least 1 alternative exon, and AS patterns are frequently tissue-biased (Pan *et al.* 2008; Wang *et al.* 2008). Tissue-specific splicing regulates many critical biological processes that include nervous system development (Raj and Blencowe 2015; Vuong *et al.* 2016), pluripotency in stem cells (Gabut *et al.* 2011), cancer (Climente-González *et al.* 2017), and neurodegenerative disorders (Porter *et al.* 2018). AS patterns are largely determined by the interaction between *cis*-elements or nucleotide sequences in the pre-mRNA and *trans*-acting factors such as RNA-binding proteins (RBPs; Wang and Burge 2008; Chen and Manley 2009; Lee and Rio 2015). Both within and between species, differential splicing patterns are predominantly driven by *cis*-element evolution (Barbosa-Morais *et al.* 2012; Wang *et al.* 2019). However, variations in the activity or abundance of RBPs and other *trans*-factors are also known to regulate tissue-specific gene expression and AS events as species diverge over longer evolutionary timescales (Irimia *et al.* 2011; Gueroussov *et al.* 2015; Torres-Méndez *et al.* 2019).

The nervous system, given its diversity of cell subtypes, has garnered much research focus as an organ system impacted by

regulation of AS (Raj and Blencowe 2015; Vuong *et al.* 2016). This cellular and splice variant diversity has been exemplified through short and long-read transcriptome profiling studies in brain regions and single cells across a variety of metazoa (Zeisel *et al.* 2015; Tasic *et al.* 2018; Furlanis *et al.* 2019; Joglekar *et al.* 2021, 2024; Patoway *et al.* 2024). However, the expression patterns of RBPs and respective AS mechanisms that drive these intricate neuron-specific transcriptomic signatures have yet to be fully characterized. Recent studies of AS regulation across transcriptionally unique neuronal cell types demonstrated that several RBPs are associated with the control of splicing programs in specific neurons, including members of the CUGBP, ELAV-like family (CELF) of regulators (Furlanis *et al.* 2019; Feng *et al.* 2021).

CELF family RBPs consist of 6 members, contain 3 well-conserved RNA recognition motifs (RRMs), and demonstrate both a nuclear and cytoplasmic distribution (Dasgupta and Ladd 2012; Peng *et al.* 2024). In mammals, CELF1-2 are generally expressed broadly across tissue types but fluctuate during development, whereas, CELF3-6 are largely expressed in the nervous system (Nasiri-Aghdam *et al.* 2021; Peng *et al.* 2024). These regulators have been implicated in various co- and post-transcriptional processes, including the regulation of AS, RNA localization, mRNA translation, and RNA stability (Dasgupta and Ladd 2012; Nasiri-Aghdam *et al.* 2021; Peng *et al.* 2024). Studies of CELF proteins in the context of splicing have identified that these factors can both activate and repress alternative exon usage in

tissue-biased manners, depending on their position of binding to target pre-mRNAs (Ladd *et al.* 2001; Kalsotra *et al.* 2008; Dembowski and Grabowski 2009; Barron *et al.* 2010; Kuroyanagi, Watanabe, and Hagiwara 2013; Kuroyanagi, Watanabe, Suzuki, *et al.* 2013). However, further study of the relationship between the positions of CELF-binding sites and splicing regulatory outcomes would provide novel insights into CELF protein-dependent regulatory mechanisms.

*Caenorhabditis elegans* is an attractive model system to study regulation by CELF proteins. There are 2 CELF paralogs in the worm, encoded by the *unc-75* and *etr-1* genes (Milne and Hodgkin 1999; Loria *et al.* 2003). ETR-1 has been implicated in muscle development, cell corpse engulfment in the germline, and neuronal migration via its activity in muscle cells (Milne and Hodgkin 1999; Boateng *et al.* 2017; Ochs *et al.* 2020, 2022). UNC-75 is broadly expressed in the nervous system and has been implicated in neuronal development, synaptic function, and the regulation of axon regeneration (Loria *et al.* 2003; Kuroyanagi, Watanabe, and Hagiwara 2013; Kuroyanagi, Watanabe, Suzuki, *et al.* 2013; Norris *et al.* 2014; Chen *et al.* 2016). Previous transcriptome-wide studies in whole animals have found that UNC-75 largely regulates the splicing of genes related to the development and physiology of the nervous system, and target splicing events are enriched for UNC-75 UG-rich consensus motifs (Kuroyanagi, Watanabe, Suzuki, *et al.* 2013; Norris *et al.* 2014; Chen *et al.* 2016; Koterniak *et al.* 2020).

In previous work, we generated tissue-enriched RNA-seq data sets to characterize tissue-specific alternative exons in *C. elegans* from 3 tissue types (neuron, muscle, and intestine) (Koterniak *et al.* 2020). We characterized hundreds of tissue- and neuron-specifc AS events, including a number of switch-like alternative exons with opposing splicing outcomes in different tissues. Among these switch-like exons is alternative exon 9 of the Zonula Occludens tight junction gene ortholog *zoo-1* (Lockwood *et al.* 2008; Koterniak *et al.* 2020). Motif analysis of *zoo-1* exon 9 and its surrounding intronic regions led to the implication of 2 intronic UNC-75 motifs overlapping with the 5′ splice site that play a key role in supporting exon 9 skipping in neurons (Koterniak *et al.* 2020).

Here, we perform a detailed characterization of the role of UNC-75 and critical *cis*-elements that dictate neuronal and nonneuronal *zoo-1* exon 9 splicing patterns. We demonstrate that UNC-75 is both necessary in neurons and sufficient in muscle cells to drive robust exon skipping. We also identify 2 sets of UNC-75 consensus sequences in both upstream and downstream introns flanking *zoo-1* exon 9 that are required for exon skipping in neurons, suggesting a mechanism involving combinatorial binding for robust switch-like behavior. Large intronic deletions disrupted splicing efficiency and neuronal splicing patterns, suggesting that longer endogenous intronic sequences are required for proper regulation of AS at this locus. Lastly, one UNC-75 consensus sequence heterologously positioned immediately downstream of an unrelated alternative exon generated *zoo-1* exon-9-like skipping patterns in neurons, further corroborating a role for 5′ splice site overlapping UNC-75 motifs as more broadly acting repressor elements. Collectively, our work provides novel mechanistic insights into how CELF proteins, an important class of RBPs, can act as potent repressors of alternative exons in a tissue-dependent context.

# Materials and methods
## *Caenorhabditis elegans* maintenance and strains used in this study
Animals were maintained at 21°C and grown on nematode growth media plates seeded with OP50-1 bacteria under standard conditions (Brenner 1974). The strain TR1331 *smg-1(r861) I* was obtained from the *Caenorhabditis* Genetics Center (CGC). All plasmids and reporters used in this study were injected into *smg-1(r861)* mutant worms to block the nonsense-mediated decay pathway, which impacts the relative abundance of our splicing reporter isoforms (Calarco and Pilaka-Akella 2022). A list of all other strains used in this study can be found in Supplementary Table 2.

## Construction of 2-color splicing reporter plasmids, microinjection, microscopy, and quantification
Two-color reporters were generated using standard molecular biology approaches and visualized by confocal microscopy and quantified by RT-PCR. Detailed methods on the construction of two-color reporters have been published (Calarco and Pilaka-Akella 2022). For a list of primers used in the study, see Supplementary Table 3. For a list of plasmids used, see Supplementary Table 4. To introduce mutations in putative UNC-75 binding sites, primers with mutated nucleotides at specified positions were used to generate PCR products, which were then stitched together by Gibson Assembly. Specifically, for region 1, the sequence TC**TT**GGTTTTTT**TT**TC was mutated to TC**CC**GGTTTTTT**CC**TC. For region 2, the sequence TG**TT**GTAACTCCG**TT**GTGT was mutated to TG**CC**GTAACTCC G**CC**GTGT. For Region 3A/B, the sequence TG**TT**GTG**TT**GTG was mutated to TG**CC**GTG**CC**GTG. See Fig. 3 for a schematic of the locations of these regions.

All microscopy was performed using a Leica SP8 laser scanning confocal. *zoo-1* exon 9 two-color reporter imaging was performed with a 40× oil immersion objective, and GFP::UNC-75 fluorescence imaging was performed with a 63× oil immersion objective.

For GFP::UNC-75 fluorescence intensity image analysis, CellProfiler (Lamprecht *et al.* 2007) was used to automatically identify and count the number of cells in each image based on GFP intensity using the IdentifyPrimaryObjects module and determine the intensity of each cell using the MeasureObjectIntensity module. The specific CellProfiler pipeline used is available upon request.

## Multiple sequence alignments and analysis of sequence conservation
For multiple sequence alignments of *zoo-1* exon 9 and its flanking sequences, we obtained *C. elegans* sequence coordinates for exon 9 and the relevant region spanning 2 introns and the neighboring upstream and downstream exons from ParaSite (Howe *et al.* 2017) and the *Caenorhabditis* Genomes Project (https://zenodo.org/records/12633738) databases. *zoo-1* orthologs were identified in 16 additional species using the program OrthoFinder (Emms and Kelly 2019). Multiple sequence alignments were generated using MUSCLE (Edgar 2004) to identify critically conserved *cis*-regulatory elements. IQ-TREE was implemented to generate a *zoo-1*-specific gene tree (Nguyen *et al.* 2015). Conserved sequences that resemble the known UNC-75 consensus sequence (Norris *et al.* 2014) were targeted for mutagenesis. The retrieval of UNC-75 protein orthologs and protein alignments (Supplementary Fig. 2) was performed in a similar manner to the *zoo-1* alignment. Notably, this resulted in direct orthologs identified for 15 out of the 17 species compared. In the case of *C. kamaaina*, the orthology finding program split the UNC-75 open reading frame into 2 distinct but well-aligning coding sequences. For *C. remanei*, although the orthology program did not find a direct match to the *C. elegans* UNC-75 protein sequence, a secondary BLAST analysis using the putative UNC-75 protein sequence from a more closely related species (*C. sp33*) as a query identified a putative ortholog. For UNC-75 RRM alignments, UNC-75 and human CELF4/5/6 protein sequences were obtained from the NCBI protein database, and each RRM domain was aligned individually using MUSCLE.

## RNA extraction, RT-PCR, gel electrophoresis

All strains were propagated by picking 3 gravid adults onto a fresh NGM plate with OP50 culture; 3 days later plates contain a mixed stage population of worms, a fraction of which express plasmids of interest as extra-chromosomal arrays. Four plates of mixed staged worms are washed with M9 solution and transferred to a 1.5 mL tube. Worm pellets were washed 3 times using M9. Total RNA was then extracted from whole animals. Hundred microliters of worm pellet was homogenized by vortexing in 400 μL of Tri-reagent (Sigma-Aldrich), snap frozen and re-thawed, followed by addition of 120 μL of chloroform (Sigma-Aldrich) to extract total RNA. The solution is centrifuged at 16,000×g, at 4°C for 15 min. The resulting upper aqueous phase was isolated, mixed with 95–100% ethanol, and total RNA was purified using the Zymo RNA Clean and Concentrator kit (R1013) as recommended by the manufacturer.

RT-PCRs were conducted by using Qiagen One-Step kit (210212). 50 ng of total RNA from each sample was used in 10 μL reaction volumes. PCRs were conducted on a Bio-Rad Thermo Cycler (1852148) for 30 cycles; the annealing temperature was set to 57°C, and extension time was 1 min. Samples were then run on a 2% agarose gel for 1 h and 30 min at 100 V. Densitometric measurements of semi-quantitative RT-PCRs were conducted using Fiji. A box was drawn over the first band of interest; all subsequent bands were measured by dragging the same box over each band, as not to change the total area measured. The raw integrated density (RawIntDen) was recorded for each region of interest, which is the sum of all the pixels in the region of interest. PSI values are then calculated by dividing the RawIntDen value of the included band by the sum of the RawIntDen values of the included and skipped bands. A Student's t-test was conducted to assess statistical significance for each experiment.

In the UNC-75 isoform overexpression experiments, transgenic animals underwent larval arrest, and this limited the material we could collect among F1 animals. Additionally, the amount of reporter amplified is a result of the number of transgenic animals present on plates at the time of collection and the relative number of cells expressing the reporters within each animal. It is thus difficult to control for overall signal generated in these experiments, and thus we simply loaded the same amounts of total RNA collected. However, PSI value measurements are internally normalized from sample to sample because the measurement calculates the percent exon inclusion within each sample. Thus, PSI values, as a measure of splicing outcome, can be directly compared between samples independent of overall signal abundance, provided they can be faithfully detected on the gel and densitometry can be performed.

## Plasmids used to express UNC-75 2RRM and 3RRM isoforms in neurons and muscle cells

The UCSC Genome Browser was used to retrieve the endogenous unc-75 nucleotide sequence. Endogenous intronic sequences were removed, and 3 artificial *C. elegans*-specific introns were added into the unc-75 coding sequence. This UNC-75 nucleotide sequence was then ordered from Twist Bioscience and cloned into a reporter downstream of a pan-neuronal promoter (Prgef-1), or body wall muscle reporter (Pmyo-3). These reporters contain a Nuclear Localization Signal (NLS), GFP and a 2A peptide linker (in this order) upstream of unc-75, followed by a tbb-2 3′ UTR. NheI and NotI flank unc-75 on the 5′ and 3′ end, respectively. Downstream of the promoter and immediately upstream of the NLS signal, there are 3 restriction sites (ClaI, EcoRI, and KpnI). All plasmid sequences are available upon request.

To generate the GFP::unc-75 fusion protein expression plasmids, primers were used to amplify the GFP region, adding a Methionine at the 5′ end of GFP and a linker sequence GGGGSAS at the 3′ end. That amplified DNA sequence was ligated into the original *Prgef-1::NLS::GFP::2A::UNC-75* expression plasmids after NheI and KpnI digestion, which removed the original NLS::GFP::2A peptide linker.

## Statistical analysis

For the analysis of our semiquantitative RT-PCR densitometry data, we performed an unpaired Student's t-test, typically comparing the distributions of PSI values between 2 different samples. Since the number of tests performed was small, we report raw rather than corrected P-values throughout the manuscript.

## Results

### Extensive skipping of a switch-like alternative exon in neurons for the zonula occludens ortholog zoo-1

To characterize neuron-specific splicing mechanisms, we mined previously generated tissue-specific mRNA profiling data (Koterniak *et al.* 2020) to identify model tissue-biased splicing events. We identified 98 switch-like splice junctions from 61 genes (mean percent spliced in difference |ΔPSI| of ≥50%) exhibiting highly differential splicing between neurons and muscle cells (Supplementary Table 1). Among the identified switch-like splicing events was alternative exon 9 from the zoo-1 locus. Our analysis indicated that zoo-1 exon 9 is largely skipped in neurons (mean PSI of 10%), whereas it is predominantly included in muscle cells (mean PSI of 90%). To further study this model splicing event in vivo, we developed a zoo-1 exon 9 minigene two-color splicing reporter (Koterniak *et al.* 2020; Fig. 1a). In this two-color reporter, the zoo-1 minigene, containing a genomic sequence spanning exon 9 and its flanking introns and exons, is cloned upstream of 2 fluorescent protein genes (enhanced Green Fluorescent Protein [EGFP] or mCherry), which are translated in 2 distinct reading frames. Alternative exon 9, which is normally frame-preserving, is engineered to contain one extra nucleotide in order to toggle between reading frames. Thus, when exon 9 is included, the mCherry-encoded reading frame is used, and when exon 9 is skipped, the reading frame encoding EGFP is utilized (Fig. 1a) (Calarco and Pilaka-Akella 2022).

We expressed our zoo-1 exon 9 two-color reporter in the body wall musculature or nervous system through the use of tissue-specific promoters (from the myo-3 and rgef-1 genes, respectively) to assess fluorescence patterns in situ and reporter mRNA splicing patterns. In agreement with our tissue-specific mRNA profiling measurements, fluorescence microscopy of animals co-expressing the zoo-1 reporter in neurons revealed predominant expression of GFP (exon skipping), while expression in muscle cells led to predominant mCherry expression (exon inclusion; Fig. 1b). To further confirm that our reporter fluorescence signals were reflecting changes at the level of AS, we performed semi-quantitative RT-PCR assays using reporter-specific primers (Fig. 1c). As expected, our RT-PCR data confirmed that zoo-1 exon 9 is largely skipped in neurons but mainly included in muscle cells (mean $PSI_{neuron} = 11.2 \pm 3.64\%$ vs mean $PSI_{muscle} = 95.4 \pm 6.35\%$, respectively, P-value = 0.0001, Student's t-test; Fig. 1c and d). Our results indicate that zoo-1 exon 9 is a *bona fide* switch-like alternative exon that is highly skipped in neurons. Moreover, our two-color reporter recapitulates the zoo-1 exon 9 endogenous splicing pattern,

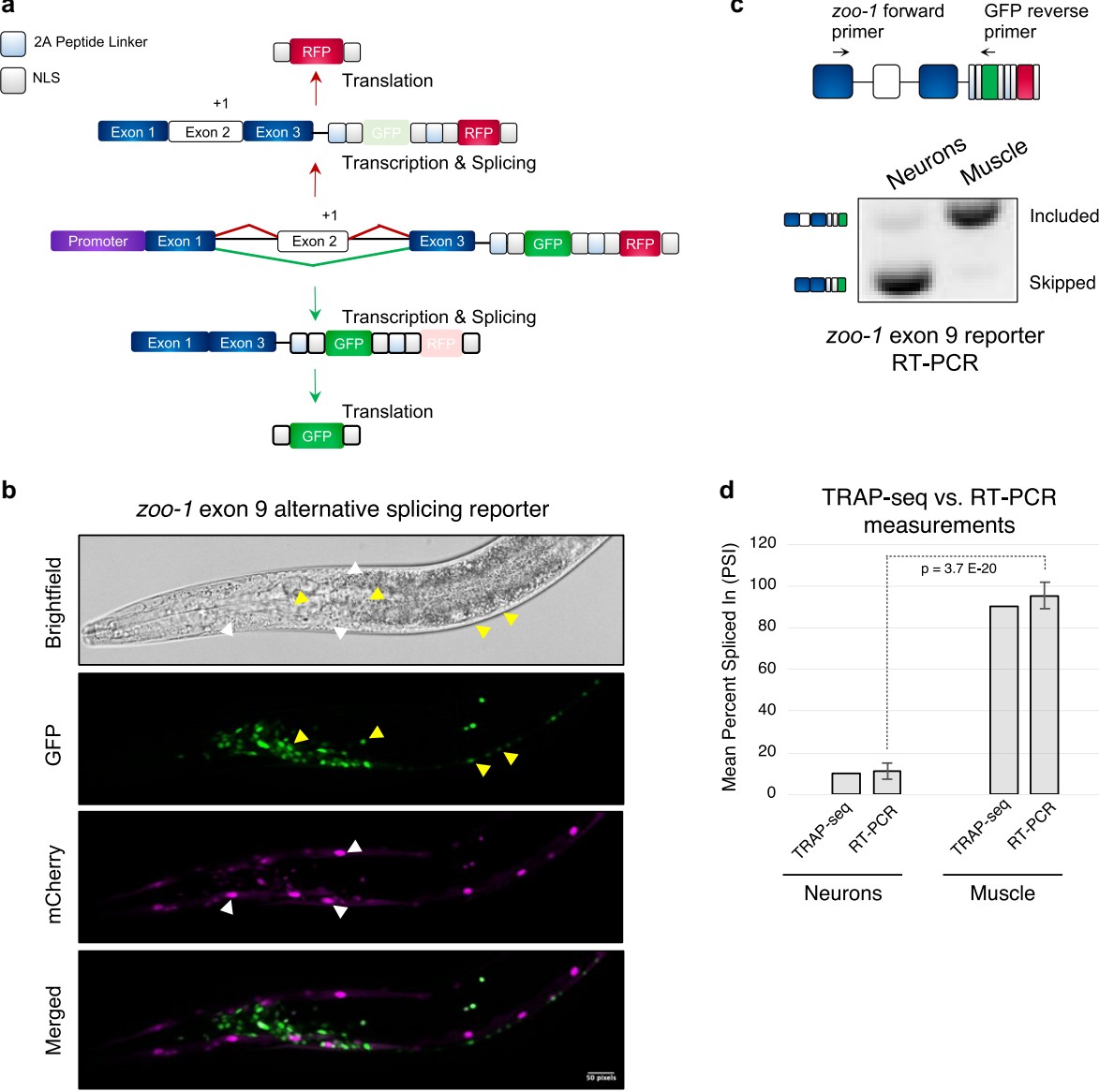

**Fig. 1.** A switch-like alternative exon in the zonula occludens ortholog zoo-1 is strongly skipped in neurons. a) Schematic of two-color splicing reporter. Reporters include the alternative exon (exon 2 in diagram) and 2 flanking introns (black lines) and exons (exons 1 and 3 in diagram). One additional nucleotide is inserted into the alternative exon, thus shifting the reading frame when included to generate mCherry, whereas when skipped the reporters generate GFP in an alternate reading frame. These reporters also contain two 2A peptide linkers and 4 NLS signals, allowing fluorescent proteins to be uncoupled from the rest of the translated peptides and localized to the nucleus. b) Fluorescence images of zoo-1 exon 9 two-color reporters expressed in neurons and muscle cells in the same animal. Representative neuronal nuclei and muscle cell nuclei are labeled with arrowheads in GFP and mCherry panels, respectively. c and d) Representative RT-PCR c) and densitometric measurements d) of zoo-1 exon 9 splicing patterns in neurons. c) The diagram shows where PCR primers anneal on the reporter and exon-included and skipped isoforms are labeled on gel. d) Densitometric measurements of RT-PCRs monitoring zoo1 exon 9 splicing in vivo and PSI values calculated from tissue-specific TRAP-seq data sets from Koterniak *et al.* (2020). $n = 3$ replicates per sample, and mean PSI ± 1 SD is plotted. P-values were calculated from a Student's *t*-test.

enabling a detailed characterization of key *cis* and *trans*-regulatory features.

## A 3 RRM-containing UNC-75/CELF protein isoform is sufficient in muscle cells for repression of zoo-1 exon 9 splicing

Our prior analysis found that UNC-75 consensus sequences are significantly enriched in neuron-biased AS events in *C. elegans* (Koterniak *et al.* 2020). Furthermore, we determined that loss of unc-75 de-represses zoo-1 exon 9 splicing in neurons, suggesting that it is normally required for skipping of this alternative exon (Koterniak *et al.* 2020). However, it is unclear if UNC-75 is sufficient

to drive skipping of this exon in nonneuronal cells. Additionally, the unc-75 gene locus encodes at least 2 protein isoforms generated through AS to contain either 2 or 3 of the highly conserved RRMs found in CELF family members (Kuroyanagi, Watanabe, and Hagiwara 2013; Fig. 2a and Supplementary Fig. 1). Thus, it is also unclear whether or not each UNC-75 protein isoform is able to drive splicing changes to a similar extent.

To more deeply explore the requirement and sufficiency of UNC-75 in regulating zoo-1 exon 9 splicing, we expressed the zoo-1 two-color reporter in neurons or muscle cells in wild-type and unc-75 knockout mutant animals, as well as in wild-type animals overexpressing UNC-75 isoforms in neurons or muscle cells in the presence

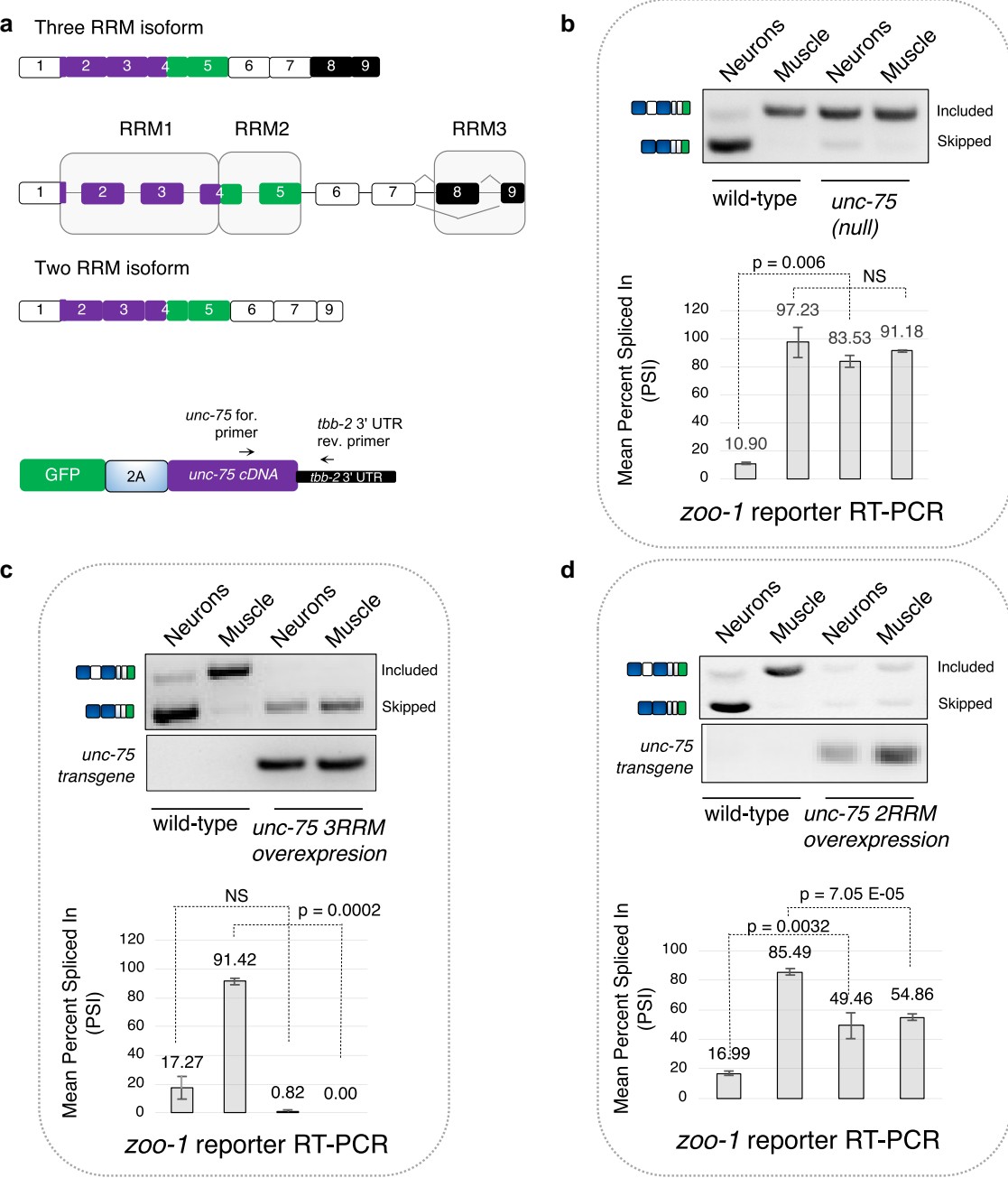

**Fig. 2.** UNC-75 is necessary in neurons and sufficient in muscle cells to elicit robust exon skipping. a) Top: schematic of UNC-75 gene architecture highlighting the different isoforms and 3 RRM domains (purple, green, and black). Exon 8 is alternatively spliced, thereby endogenously generating a 3 RRM and 2 RRM isoform. Bottom: schematic of GFP::2A peptide::unc-75 cDNA::tbb-2 3' UTR transgene used in c and d). b–d) Representative RT-PCR and densitometric measurements of zoo-1 exon 9 splicing reporter (top gels) or RT-PCR of UNC-75 transgenes (bottom gels) in neurons or muscle cells as labeled. b) Reporter splicing patterns are measured in wild-type worms or unc-75 null mutants. c) and d) Reporter splicing patterns are compared between wild-type animals and animals overexpressing the UNC-75 3 RRM-containing isoform c) or 2 RRM-containing isoform d) in neurons or muscle cells. $n = 3$ replicates for each data point, while $n = 2$ replicates for data points in c) and d). Mean PSI ± 1 SD is plotted, and P-values are calculated from Student's t-test.

of endogenous UNC-75 (Fig. 2b–d). In agreement with our previous findings, loss of UNC-75 led to a pronounced increase in zoo-1 exon 9 inclusion in neurons (Fig. 2b; wild-type mean $PSI_{neuron} = 10.89 \pm 0.89\%$, unc-75 mutant mean $PSI_{neuron} = 83.5 \pm 4.09\%$, P-value = 0.006, Student's t-test). Intriguingly, mis-expression of the UNC-75 isoform with 3 RRMs in muscle cells was sufficient to induce robust skipping of zoo-1 exon 9 in muscle, recapitulating the neuronal zoo-1 splicing pattern (Fig. 2c; wild-type mean $PSI_{muscle} = 91.4 \pm 2.11\%$, UNC-75 3 RRM overexpression mean $PSI_{muscle} = 0 \pm 0\%$, P-value =

0.0002, Student's t-test). However, while mis-expressing the UNC-75 isoform with 2 RRMs in muscle cells also resulted in significantly more skipping of zoo-1 exon 9 in muscle compared with wild-type animals (Fig. 2d; wild-type mean $PSI_{muscle} = 85.49 \pm 2.2\%$, UNC-75 2 RRM overexpression mean $PSI_{muscle} = 54.85 \pm 2.2\%$, P-value = $7.05 \times 10^{-5}$, Student's t-test), this repressive effect was weaker when compared with the effect caused by mis-expression of isoform containing 3 RRMs (compare Fig. 2c and d). Surprisingly, overexpression in neurons of the isoform containing 2 RRMs led to

increased inclusion of alternative exon 9 in neurons (Fig. 2d; wild-type mean $PSI_{neuron}$ = 16.99 ± 1.68%, UNC-75 2 RRM overexpression mean $PSI_{neuron}$ = 49.46 ± 8.73%, P-value = 0.003, Student's t-test).

It is possible that the weaker repressive effect mediated by the UNC-75 2 RRM-containing isoform could be the result of reduced protein stability and/or altered subcellular localization rather than intrinsic protein activity itself. To address these possibilities, we generated new GFP::UNC-75 2 RRM and 3 RRM fusion expression transgenes and expressed these fusion proteins in neurons. After injecting these fusion vectors at similar concentrations, we quantified protein expression and also assessed their subcellular localization by confocal microscopy (Supplementary Fig. 2). We observed that both the 2 RRM and 3 RRM protein isoforms were expressed at similar levels across animals, and both isoforms also localized to the nucleus, with an enrichment of signal in previously reported dynamic foci (Loria *et al.* 2003; Supplementary Fig. 2).

Taken together, our results suggest that each UNC-75 isoform exerts different repressive activity, and that when present at high enough levels, the isoform encoding 2 RRMs can act in a dominant negative manner to dampen the effects of the isoform encoding 3 RRMs. More broadly, our results indicate that UNC-75 is both necessary in neurons and sufficient in muscle cells to drive robust repression of *zoo-1* alternative exon 9.

## Conserved intronic UNC-75 binding sites flank the 3′ and 5′ splice sites surrounding zoo-1 alternative exon 9

Given the role of UNC-75 in regulating *zoo-1* exon 9 splicing in neurons, we aimed to identify candidate *cis*-regulatory elements bound by UNC-75 that could mediate repression of the alternative exon. Critical intronic *cis*-elements regulating AS undergo increased negative selection pressure, likely to preserve splicing patterns across nematode species (Kabat *et al.* 2006; Koterniak *et al.* 2020). We thus generated multiple sequence alignments of *zoo-1* exon 9 and its 2 flanking introns using homologous sequences from 17 *Caenorhabditis* species (part of the broader Elegans supergroup; Fig. 3). These multiple sequence alignments identified several occurrences of the core UNC-75 consensus sequences (UUGU, UGUUG, and UUGUG) to be conserved upstream and in the vicinity of the 3′ splice site flanking exon 9 (Fig. 3, region 2). Expanding on previous alignments with fewer species (Koterniak *et al.* 2020), our new analysis also highlights the extensive conservation of 2 UNC-75 consensus sequences overlapping with the 5′ splice site flanking exon 9 (Fig. 3, region 3A/3B). We additionally identified a region containing sporadic occurrences of UG-rich sequences that were not well-conserved further upstream of the 3′ splice site (Fig. 3, region 1). All of these regions were selected for mutagenesis to screen for possible impacts on exon 9 AS.

Consistent with a deeply conserved role for UNC-75-mediated regulation of *zoo-1* exon 9 splicing patterns, using the *C. elegans* UNC-75 protein sequence as a query, we identified clear *unc-75* orthologs in 15 out of 17 species we queried above for our comparative analysis of *zoo-1* sequences (Supplementary Fig. 3). In the case of *C. kamaaina*, the orthology finding program split the UNC-75 open reading frame into 2 distinct but well-aligning coding sequences. For *C. remanei*, although the orthology program did not find a direct match to the *C. elegans* UNC-75 protein sequence, a secondary BLAST analysis using the putative UNC-75 protein sequence from a more closely related species (*C. sp33*) as a query identified a putative ortholog. Taken together, these data suggest that UNC-75-mediated repression of *zoo-1* alternative exon 9 is an ancient mechanism that evolved in *Caenorhabditis* nematodes.

## UNC-75 consensus UG-rich elements in intronic regions flanking both sides of zoo-1 exon 9 are required for exon skipping in neurons

We generated a series of point mutations disrupting the 3 potential UNC-75-binding regions identified above. First, we investigated whether the 2 UG-rich elements that overlap with the 5′ splice site (designated as regions 3A and 3B) are critical for *zoo-1* exon 9 splicing (Figs. 3 and 4a). Point mutations disrupting region 3A alone resulted in a mild but significant stimulation of exon 9 splicing. However, analogous mutations in region 3B alone had no significant impact on splicing patterns in neurons (Fig. 4a). However, disrupting both region 3A and 3B together resulted in a synergistic de-repression of exon 9 in neurons (Fig. 4a; wild-type mean $PSI_{neuron}$ = 13.86 ± 1.0%, region 3A/3B mutant mean $PSI_{neuron}$ = 40.11 ± 4.1%, P-value = 0.0004, Student's t-test) but had minimal effects on *zoo-1* exon 9 splicing in muscle cells (Fig. 4a). Consistent with our previous findings (Koterniak *et al.* 2020), these data suggest that the 2 *cis*-regulatory elements downstream of the 5′ splice site work together to play a critical role in regulating *zoo-1* exon 9 splicing in neurons but not in muscle cells.

Next, we investigated the potential *cis*-regulatory effects of regions 1 and 2. Point mutations disrupting each region individually revealed a robust de-repression of *zoo-1* exon 9 when region 2 is mutated (Fig. 4b; wild-type mean $PSI_{neuron}$ = 12.0 ± 1.0%, region 2 mutant mean $PSI_{neuron}$ = 68.0 ± 3.9%, P-value = 2.54 × 10$^{-5}$, Student's t-test) but no impact on splicing patterns when region 1 was mutated (Fig. 4b). Mutation of both regions 1 and 2 simultaneously also resulted in a strong increase in exon 9 inclusion in neurons, but this effect was similar to what was observed in reporters with region 2 alone mutated (Fig. 4b).

Given the observed repressive activity of regions 2 and 3A/3B, we next tested whether the combined mutation of all of these *cis*-elements led to an additive effect on *zoo-1* exon 9 inclusion. Indeed, when this combinatorial mutant reporter was expressed in neurons, a further increase in exon inclusion was observed compared with mutations of either region 2 or 3A/3B alone, more closely resembling what was observed in neurons in an *unc-75* null mutant background (Supplementary Fig. 4; wild-type mean $PSI_{neuron}$ = 4.0 ± 1.8%, region 2/3A/3B mutant mean $PSI_{neuron}$ = 81.7 ± 3.7%, P-value = 3.9 × 10$^{-6}$, Student's t-test; compare with Fig. 2b).

Our previous work has shown that tissue-biased alternative exons in *C. elegans* are flanked on average by longer introns (median size = 172 nucleotides), while constitutively spliced exons have shorter flanking introns (median size = 58; Koterniak *et al.* 2020). Having identified 2 critical *cis*-regulatory elements, each located within 15 nucleotides of the 3′ and 5′ splice sites of *zoo-1* exon 9, we next tested whether full-length intronic regions are required or whether minimal sequences containing core splicing signals and regions 2 and 3 are sufficient, for exon 9 splicing in neurons and muscle cells. The upstream and downstream *zoo-1* introns 8 and 9 are 456 nucleotides and 959 nucleotides long, respectively. We generated reporters deleting most of these intronic regions, with the exception of 30 nucleotides adjacent to each of the 4 splice sites, in an effort to conserve core splicing signals and critical regions 2 and 3 within *zoo-1* introns 8 and 9 (Supplementary Fig. 5).

Deletion of large portions of upstream and downstream introns independently led to a significant de-repression of exon 9 splicing in neurons, with loss of intron 9 sequence having a more pronounced effect than intron 8 (Supplementary Fig. 5; wild-type mean $PSI_{neuron}$ = 11.6 ± 4.4%, intron 9 deletion mean $PSI_{neuron}$ =

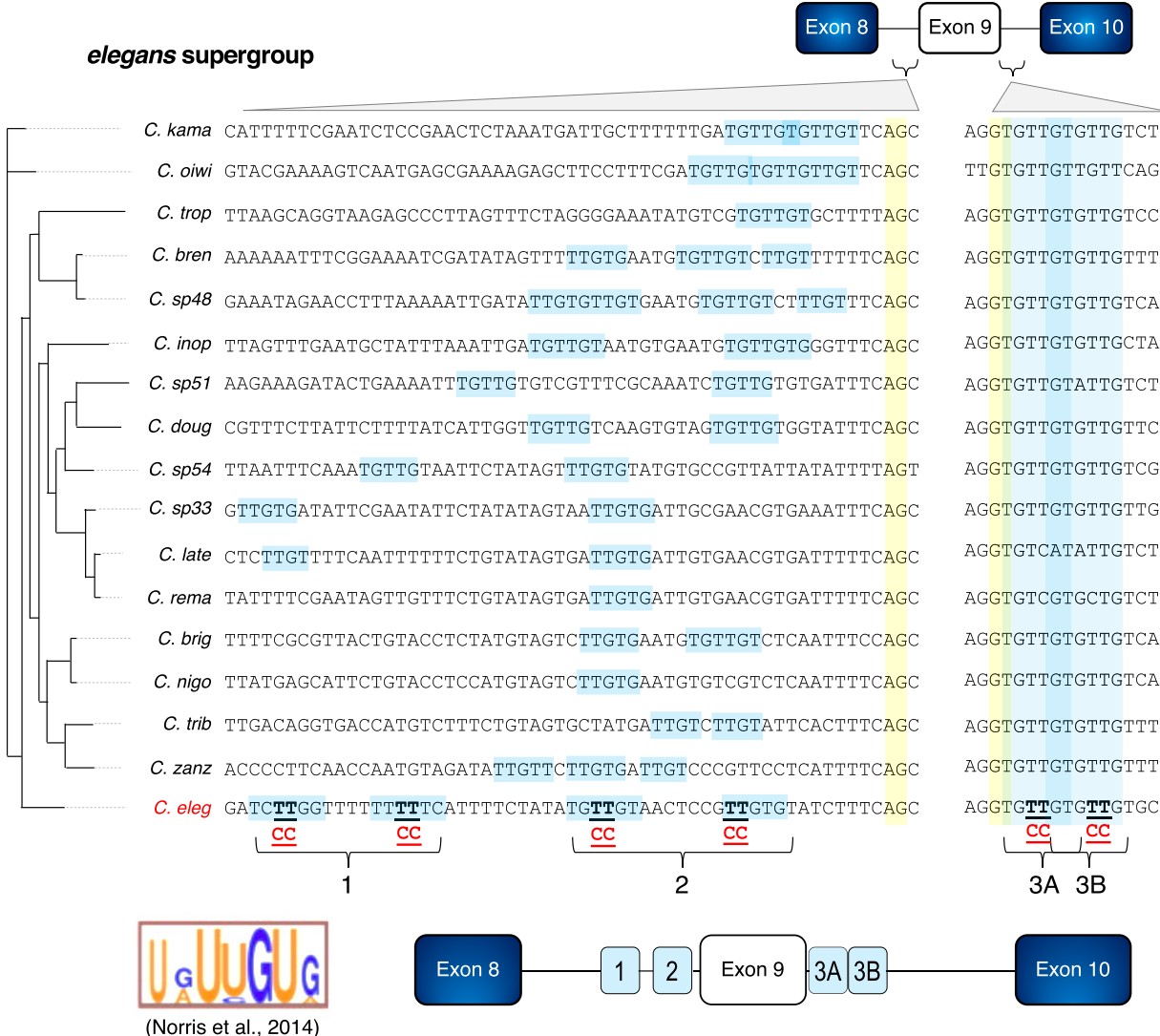

**Fig. 3.** Conserved intronic UNC-75 binding sites flank the 3′ and 5′ splice sites surrounding zoo-1 alternative exon 9. zoo-1 ortholog sequence alignments were obtained from 17 *Caenorhabditis* species in the *elegans* super group. Sixty nucleotides upstream of exon 9 (including the 3′ splice site) and 12 nucleotides downstream of the 5′ splice site are shown. Canonical splice sites are highlighted in yellow, while sequences matching the UNC-75 consensus motif are highlighted in light blue. TT dinucleotides in bold and underlined are the nucleotides mutated (to CC) for regions 1, 2, 3A, and 3B, as tested in Fig. 4. The UNC-75 consensus binding motif identified by RNACompete is displayed (bottom left) for reference (Norris *et al.* 2014). Bottom right: schematic of 3 potential *cis*-regulatory elements (light blue boxes; regions 1, 2, and 3A/3B as referenced in text) identified by multiple species alignments and tested in Fig. 4. Species abbreviations: *kama, kamaaina; oiwi, oiwi; trop, tropicalis; bren, brenneri; inop, inopinata; doug, doughertyi; late, latens; rema, remanei; brig, briggsae; nigo—nigoni; trib, tribulationis; zanz, Zanzibari; eleg, elegans.*

28.3 ± 4.4%, P-value vs wild-type = 0.0001, Student's *t*-test; intron 8 deletion mean $PSI_{neuron}$ = 46.8 ± 1.07%, P-value vs. wild-type = $1.77 \times 10^{-8}$, Student's *t*-test). Interestingly, deletion of both intronic regions further increased exon 9 inclusion in neurons, but also led to substantial accumulation of fully unspliced reporter pre-mRNA in both neuronal and muscle cells (Supplementary Fig. 5). These results suggest that either longer intronic sequences per se or additional *cis*-regulatory features contained in these deleted intronic regions, or both, are needed for proper tissue-specific splicing regulation of this exon.

Collectively, these results identify 2 distinct regions with conserved, strong UNC-75 consensus motifs located in the vicinity of the splice sites flanking zoo-1 alternative exon 9 as the most critical regulators of this AS pattern. Our results also reveal an important influence of other intron characteristics to enable proper tissue-biased regulation of this exon.

## 5′ splice site-adjacent UNC-75 consensus sequences are sufficient to increase exon skipping in neurons, likely by weakening the 5′ splice site

UNC-75 consensus sequences located adjacent to the 5′ splice site play a critical role in zoo-1 exon 9 skipping in neurons. To investigate whether this mechanism is sufficient to more generally drive exon skipping patterns in neurons, we manipulated a different splicing reporter previously used in our lab to monitor exon 16 of the unc-16 locus (Norris *et al.* 2014). Exon 16 of unc-16 is alternatively spliced, and its downstream intron contains consensus binding sites for UNC-75 and another RBP (EXC-7) located ~30 nucleotides downstream of the 5′ splice site (Fig. 5a). Our previous study demonstrated that UNC-75 and EXC-7 function together by recognizing multiple *cis*-elements to stimulate inclusion of this exon in the nervous system (Norris *et al.* 2014). We thus reasoned that bringing the intronic UNC-75 consensus binding motif

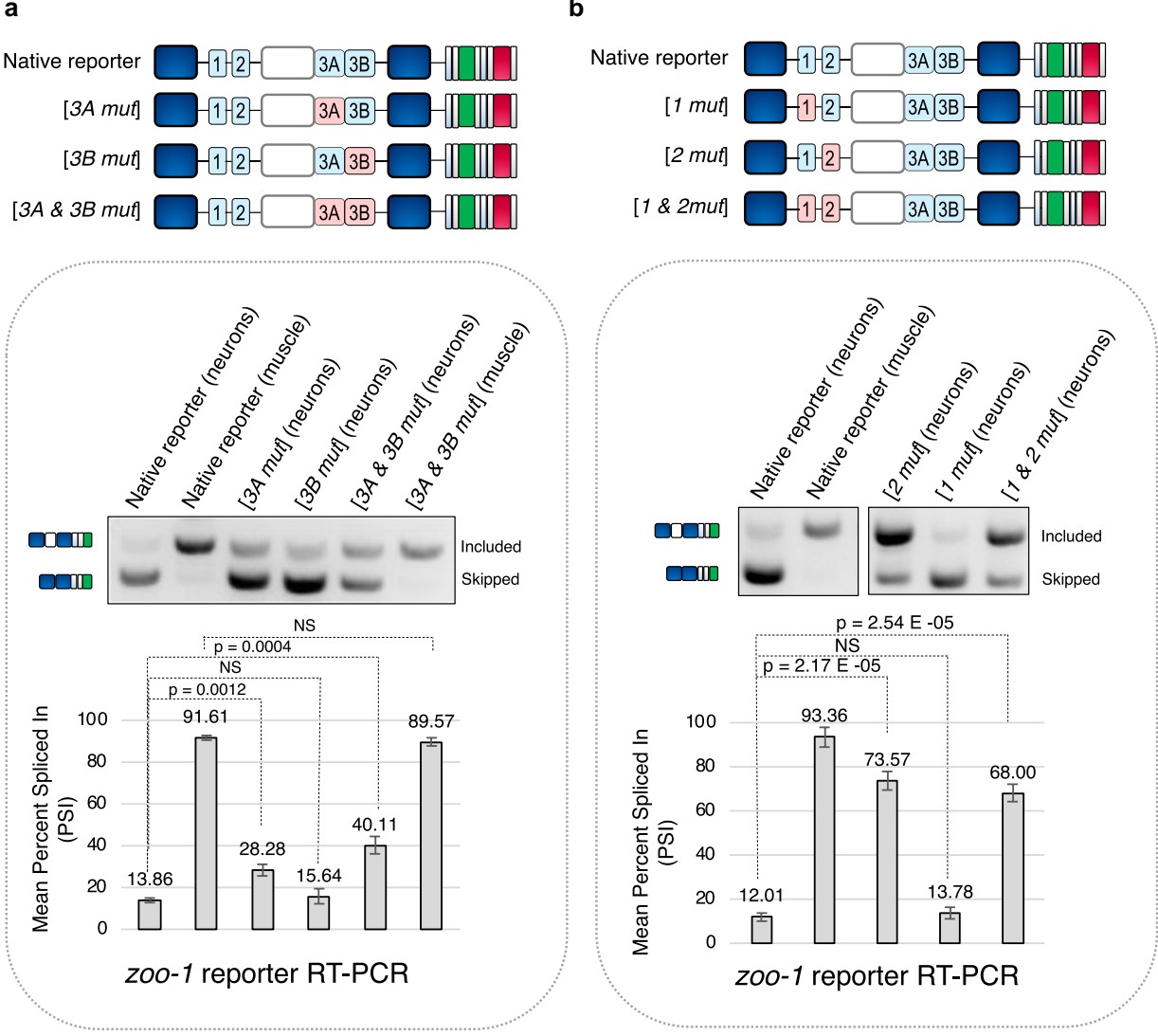

**Fig. 4.** UNC-75 consensus UG-rich elements in intronic regions flanking both sides of zoo-1 exon 9 are required for exon skipping in neurons. a and b) Top: schematics of mutagenesis experiments disrupting 2 UNC-75 consensus sequences immediately adjacent to the 5′ splice site flanking exon 9 and 2 regions upstream of exon 9. Pink labeled boxes denote regions mutated as shown in Fig. 3. Bottom: representative RT-PCR and densitometric measurements assessing zoo-1 exon 9 reporter splicing patterns in neurons and muscle cells, with each cis-element mutation labeled accordingly. $n = 3$ replicates for each data point. Mean PSI ± 1 SD is plotted, and P-values are calculated from Student's t-test.

closer to the 5′ splice site-adjacent to exon 16, while preserving the overall intron length, the native flanking sequences, and the relative distance of the EXC-7 motif to the 5′ splice site (Fig. 5a)—would lead to more exon skipping.

Interestingly, positioning the UNC-75 consensus sequence immediately adjacent to the 5′ splice site led to a significant increase of exon 16 skipping (Fig. 5b; wild-type mean $PSI_{neuron} = 34.0 \pm 6.2\%$, UNC-75 site at intron +1 $PSI_{neuron} = 11.9 \pm 6.9\%$, P-value = 0.04, Student's t-test). This repressive effect was very position-specific, since placing the consensus sequence further downstream of the 5′ splice site (3–7 nucleotides from the GU dinucleotide) did not lead to pronounced exon skipping and, in some cases, even induced a modest stimulatory effect (Fig. 5b). Moreover, mutation of the intronic UNC-75 binding site altogether had no impact on alternative exon inclusion (Fig. 5b), further indicating that the repressive effect we observed was likely due to positioning the motif over the 5′ splice site rather than via a loss of activity from its endogenous position. However, the current results do not rule out

that moving the UNC-75 motif further away from its neighboring EXC-7 binding site could also influence exon inclusion.

Positioning the UNC-75 motif over the 5′ splice site generates a weak splice site that deviates from a strong consensus sequence match. Thus, repressive effects observed in the +1 intron motif mutant reporter could be the result of creating weaker recognition of the flanking 5′ splice site rather than due to UNC-75-mediated repression. To address this possibility, we tested the +1 intron motif mutant reporter in an unc-75 mutant background (Fig. 5c). Consistent with our previous work (Norris et al. 2014), loss of unc-75 resulted in an increase in exon skipping of the native reporter (Fig. 5c; native reporter in wild-type mean $PSI_{neuron} = 42.8 \pm 6.6\%$, native reporter in unc-75 null $PSI_{neuron} = 27.0 \pm 5.3\%$, P-value = 0.03, Student's t-test). However, this repressive effect was not as strong as the skipping observed in the +1 intron motif mutant reporter. Moreover, we found that loss of UNC-75 did not lead to derepression of exon 16 skipping in the +1 intron motif mutant reporter when compared with a genetic background containing

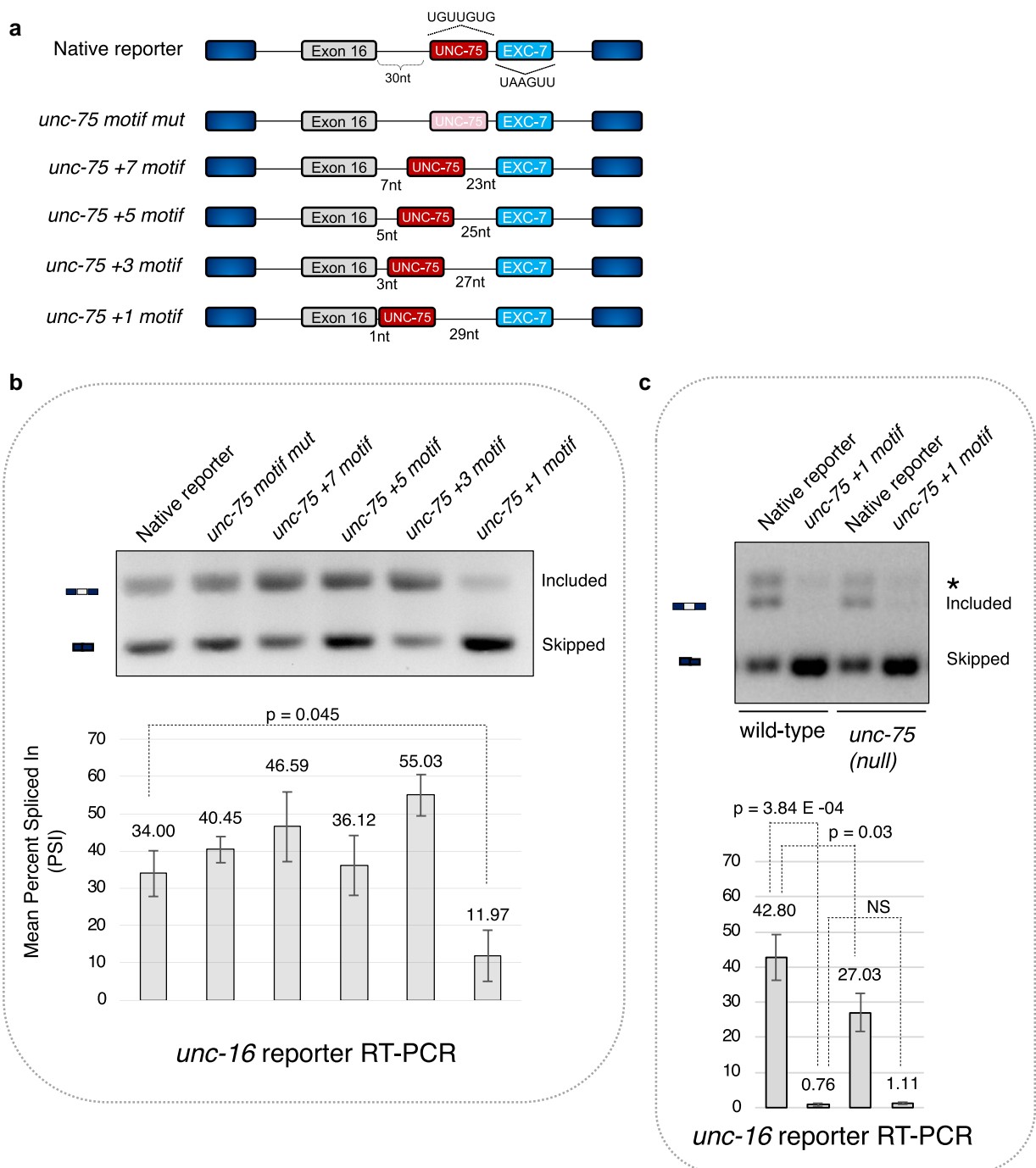

**Fig. 5.** 5′ splice site-adjacent UNC-75 consensus sequences are sufficient to increase exon skipping in neurons. a) Schematic of *unc-16* alternative exon 16 mutagenesis experiment, displaying the native reporter, a reporter with the intronic UNC-75-binding motif mutated (UGUUGUG to UGCCGUG; *motif mut*), and a series of reporters shifting the endogenous UNC-75 consensus sequence closer to the 5′ splice site (from positions +7 to +1) while preserving the overall intron size, endogenous flanking sequences, and maintaining the position of the neighboring EXC-7-binding site. b) Representative RT-PCR, and densitometric measurements of the *unc-16* reporters. c) Representative RT-PCR, and densitometric measurements of the native and the +1 motif mutant *unc-16* reporters expressed in wild-type or *unc-75* null mutant genetic backgrounds. Asterisk denotes likely heteroduplex species. $n = 3$ replicates for each data point. Mean PSI $\pm$ 1 SD is plotted, and P-values are calculated from student's *t*-test.

functional UNC-75 (Fig. 5c; *unc-75* site at intron +1 in wild-type background PSI$_{neuron}$ = 0.8 $\pm$ 0.4%, *unc-75* site at intron +1 PSI$_{neuron}$ = 1.1 $\pm$ 0.4%, P-value = 0.35, Student's *t*-test).

Collectively, our data demonstrate that an UNC-75 consensus sequence immediately overlapping a 5′ splice site is sufficient to increase exon skipping in a heterologous context. However, our data suggest that in this context, repression is likely mediated through weakening the 5′ splice site rather than through direct repression by UNC-75.

## Discussion

### Multiple flanking intronic cis elements ensure robust CELF-mediated exon skipping

In this study, we characterized the role of the CELF protein UNC-75 in the skipping of a model switch-like alternative exon (*zoo-1* exon 9) in neurons. Collectively, our experiments support a model where UNC-75 relies on 2 key intronic *cis* elements—one situated

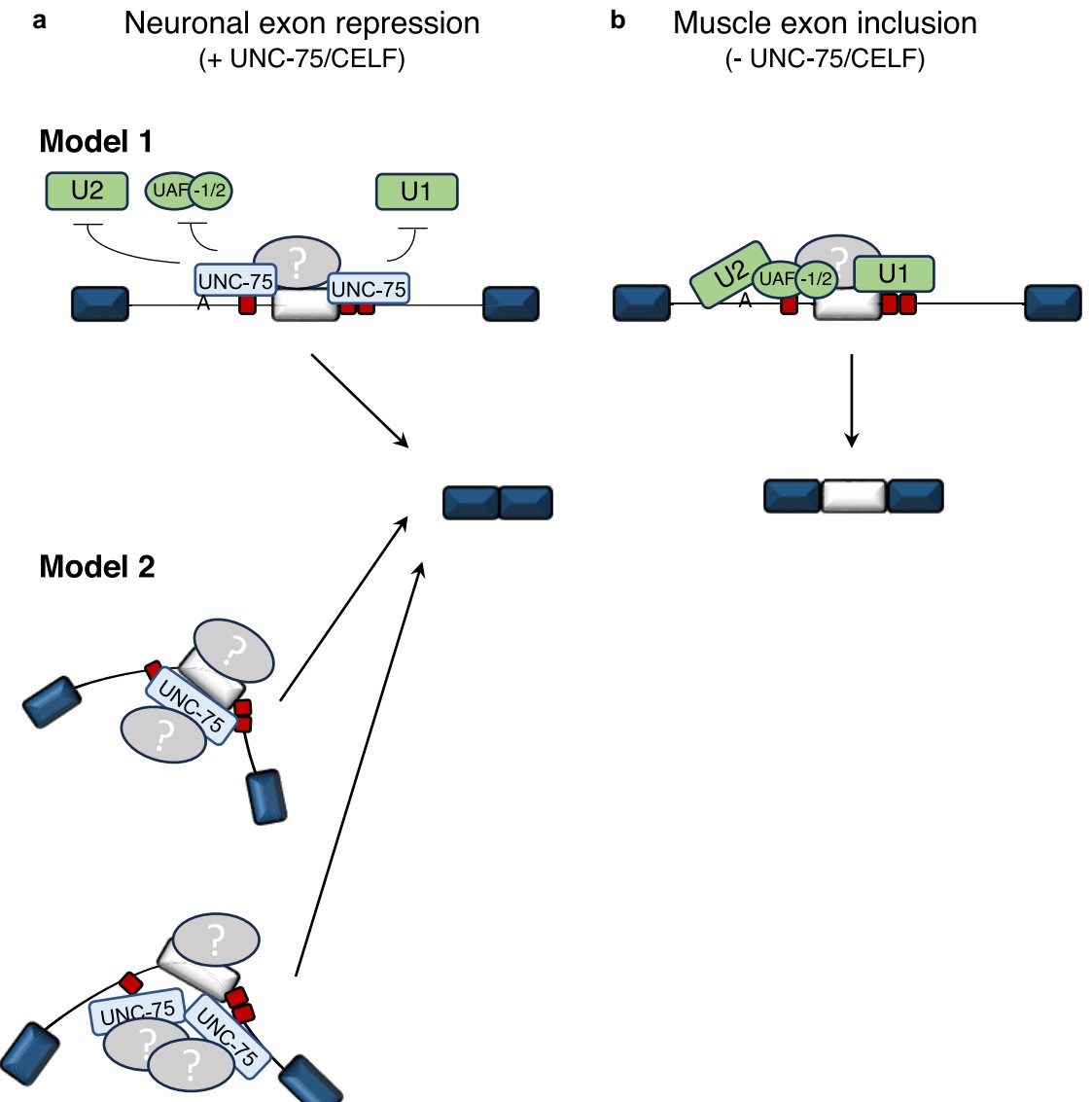

**Fig. 6.** Model for UNC-75 mediated switch-like regulation of zoo-1 exon 9 in neurons and nonneuronal cells. a) Schematic of proposed mechanisms of zoo-1 exon 9 splicing in neurons. In model 1, binding of UNC-75 to upstream and downstream intronic regions prevents the binding and/or recruitment of the U1 snRNP, U2AF complex (UAF-1/2) and the U2 snRNP. In this model, multiple independent binding events create a robust repression of exon usage in neurons. In model 2, which may not be mutually exclusive, one or more UNC-75 proteins may interact with both upstream and downstream intronic-binding sites to create a looped conformation. b) Schematic of proposed mechanism of zoo-1 exon 9 splicing in muscle cells, where in the absence of UNC-75 the alternative exon is highly included, likely through an exon definition mechanism. In all diagrams, yet to be identified cofactors are included (gray ovals), which likely facilitate the switch-like behavior in each cell type.

upstream of exon 9, and the other overlapping the downstream 5′ splice site, to achieve robust exon skipping in neurons (Fig. 6). Given proximity or overlap of these *cis* elements to core splicing signals, we speculate that in neurons the presence of UNC-75 binding would lead to steric interference of key factors involved in early spliceosome formation, such as SFA-1/SF1 or the U2 snRNP at the branchpoint, the UAF-1/2 (U2AF) heterodimer at the 3′ splice site, and the U1 snRNP at the 5′ splice site (Fig. 6). Interestingly, such mechanisms interfering with recognition of the branch point or 3′ splice site have been suggested for specific alternative exons repressed by CELF proteins (Dembowski and Grabowski 2009; Barron *et al.* 2010).

The results of our mutagenesis of individual and multiple combinations of binding sites (Fig. 4 and Supplementary Fig. 4) suggest an additive role for these *cis* elements in ensuring near-complete exon skipping in neurons, highlighting the importance of multiple binding sites influencing control by a single factor to create robust splicing outcomes (Chen and Manley 2009; Lee and Rio 2015). This additive effect is similar to results observed with the Rbfox, PTBP, and Nova families of splicing regulatory proteins, where the activity of these factors can be influenced by the clustering and positioning of their cognate-binding motifs (Jensen *et al.* 2000; Amir-Ahmady *et al.* 2005; Jelen *et al.* 2007; Xue *et al.* 2009; Begg *et al.* 2020). In the case of studies of CELF proteins, multiple clustered binding sites in the downstream intron flanking the chicken cardiac troponin T alternative exon are necessary to promote CELF-dependent exon inclusion (Cooper 1998; Ladd *et al.* 2001; Han and Cooper 2005). We hypothesize that multiple UNC-75 proteins engage with the upstream and downstream *cis* elements simultaneously or in sequence (Fig. 6, model 1), although our

data would also be consistent with 1 UNC-75 protein coordinating binding at both upstream and downstream intronic sequences via its multiple RRMs (Fig. 6, model 2).

This latter model would be reminiscent of how other RBPs with multiple RRM domains, such as PTBP1, can facilitate long-range RNA looping interactions either via dimerization (Xue *et al.* 2009; Ye *et al.* 2023) or via the same molecule using RRMs 3 and 4 (Amir-Ahmady *et al.* 2005; Oberstrass *et al.* 2005; Lamichhane *et al.* 2010). The weaker activity we and others have observed exhibited by UNC-75 proteins containing only 2 of 3 RRM domains (Fig. 2; Kuroyanagi, Watanabe, and Hagiwara 2013) would be consistent with a requirement for multiple RNA recognition interfaces to act simultaneously to achieve robust regulatory effects.

Additionally, altering the ratios of 2RRM/3RRM-containing isoforms across neurons would be predicted to impact splicing regulation of target pre-mRNAs, leading to neuron subtype-specific splicing patterns. In support of this concept, recent studies analyzing the transcriptomes of a large number of *C. elegans* neuron types reveal extensive neuron subtype-specific splicing patterns, and have specifically identified differential *unc-75* isoform usage across different classes of neurons (Weinreb *et al.* 2024; Wolfe *et al.* 2024). Taken together, either of the models presented above would represent a novel mechanism by which CELF proteins robustly inhibit splicing by targeting both 3′ and 5′ intronic regions flanking the same alternative exon, and warrant further exploration.

### A flexible 5′ splice site contributes to switch-like splicing control by CELF proteins

The relative recognition of competing 5′ splice sites plays a central role in contributing to AS outcomes (Roca *et al.* 2013; Malard *et al.* 2022). Here, we have identified a flexible UG-rich 5′ splice site flanking an alternative exon that likely evolved to be co-opted for regulation by UNC-75. The nearly identical AG/GUGUUG sequence conservation of the *zoo-1* exon 9 flanking 5′ splice site across *Caenorhabditis* species (Fig. 3) suggests that this particular sequence is critical for splicing regulation at this exon. We speculate that this 5′ splice site enables repression of splicing when bound by UNC-75 in neurons but remains a strong enough splice site to be recognized by the U1 snRNP in nonneuronal cells where UNC-75 is absent.

Although this 5′ splice site sequence does not match the typical CAG/GUAAGU consensus, prior work demonstrated how a diversity of variant splice sites can get recognized, in some cases, through bulged-out nucleotides and alternative base pairing registers between U1 snRNA and the pre-mRNA (Sheth *et al.* 2006; Roca *et al.* 2012; Wong *et al.* 2018). Additionally, there may be additional factors such as splicing regulatory proteins binding to other *cis* elements or RNA secondary structure acting to create a permissive environment for robust recognition of the 5′ splice site and exon inclusion in the absence of UNC-75 (Roca *et al.* 2013; Malard *et al.* 2022). This latter scenario would be consistent with our data indicating additional sequence motifs in the introns, or perhaps intron length-dependent folding of RNA secondary structures, are required for efficient splicing of the *zoo-1* minigene reporter (Supplementary Fig. 5). In contrast, although overlapping an UNC-75 consensus sequence with a heterologous 5′ splice site (flanking *unc-16* exon 16) led to exon skipping, this repressive effect was independent of UNC-75 activity and likely due to weakening the 5′ splice site (Fig. 5). These results suggest that sequence context surrounding the 5′ splice site matters at each gene locus in order to determine UNC-75/CELF-binding activity and repression. It will be interesting to determine the sequence variation tolerated at the 5′ splice site flanking *zoo-1* exon 9 that preserves UNC-75 mediated switch-like behavior. More broadly, it will be

intriguing to determine whether similar CELF-dependent 5′ splice sites have evolved in other metazoa, including mammals.

### A master regulator of the neuronal AS program in *C. elegans*?

UNC-75 mis-expression in muscle cells is sufficient to induce a neuronal switch-like AS pattern in *zoo-1* alternative exon 9 (Fig. 2). Interestingly, in our experiments, overexpression of UNC-75 in muscle cells or in the nervous system leads to developmental arrest (data not shown), suggesting that UNC-75 levels must be tightly controlled and remain specific to neurons. Consistent with this latter point, UNC-75 itself is alternatively spliced, creating protein variants with distinct numbers of RRMs and regulatory potential (Kuroyanagi, Watanabe, and Hagiwara 2013; Kuroyanagi, Watanabe, Suzuki, *et al.* 2013; Chen *et al.* 2016; Fig. 2). The *unc-75* locus was the only gene encoding a sequence-specific RBP that was uncovered in the original screens looking for uncoordinated mutants (Brenner 1974). Moreover, UNC-75 is the only RBP with expression that is specific to the nervous system. Finally, *unc-75* alleles have been recovered in more recent screens identifying regulators of neuronal AS outcomes (Norris *et al.* 2014; Chen *et al.* 2016). In light of these observations, we speculate that UNC-75 has evolved to act as a master regulator of the neuronal splicing program in nematodes. The neuronal program of isoforms regulated by UNC-75 would be reminiscent of the battery of genes turned on by terminal selector transcription factors to establish neuronal identity and neuron subtype specification (Hobert 2008; Hobert and Kratsios 2019). It will be interesting to assess the extent to which expression of UNC-75 can induce a neuronal splicing program in nonneuronal cells, akin to cellular reprogramming or trans-differentiation phenomena observed for mis-expression of chromatin regulators and transcription factors (Tursun *et al.* 2011; Zuryn *et al.* 2014; Rothman and Jarriault 2019; Marchal and Tursun 2021).

## Data availability

Mining of switch-like exons was performed using previous TRAP-seq data generated by our lab (Koterniak *et al.* 2020). Raw data from this study are available at the NCBI Gene Expression Omnibus repository under accession GSE106374.

Supplemental material available at GENETICS online.

## Acknowledgments

We thank members of the Calarco Lab for detailed discussions surrounding the manuscript. Some strains were provided by the *Caenorhabditis* Genetics Center (CGC), which is funded by NIH Office of Research Infrastructure Programs (P40 OD010440).

## Funding

P.P.-A is supported by a Doctoral Canada Graduate Scholarship from the Natural Sciences and Engineering Research Council of Canada (NSERC), and this work was supported by NSERC (Discovery Grant RGPIN-2017-06573) and the Canadian Institutes of Health Research (Project Grants 156300 and 180365) grants to J.A.C.

## Conflicts of interest

The author(s) declare no conflict of interest.

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

*Editor: C. Phillips*