## [Peer Review File · Genetics]

Neuron-specific repression of alternative splicing by the conserved CELF protein UNC-75 in *C. elegans*

Pallavi Pilaka-Akella, Nour Sadek, Daniel Fusca, Asher Cutter, and John Calarco

NOTE: The reviews and decision letters are unedited and appear as submitted by the reviewers.

In extremely rare instances and as determined by a Senior Editor or the EIC, portions of a review may be redacted. If a review is signed, the reviewer has agreed to no longer remain anonymous.

The review history appears in chronological order.

Review Timeline:

Submission Date:	2024-09-24
Editorial Decision:	2024-10-18
Revision Received:	2025-01-28
Accepted:	2025-01-29

October 18, 2024

RE: GENETICS-2024-307490

Dear Dr. Calarco:

I am pleased to accept your manuscript entitled "Neuron-specific repression of alternative splicing by the conserved CELF protein UNC-75 in *C. elegans*" for publication in GENETICS, pending minor revision.

Please submit your revision along with a brief description of how you modified the manuscript in response to the reviewers' concerns and suggestions (which can be viewed at the bottom of this email. Most important are 1) addressing the reason for the lower amount of PCR product recovered in the *unc-75* 2RRM over expression animals and how it was ensured that the results are comparable (Reviewer 2, point #5), 2) showing the sub cellular localization and RNA/protein levels of the two UNC-75 isoforms (Reviewer 3, point #1), 3) evaluating whether disrupting both motifs 2 and 3 simultaneously would recapitulate the splicing phenotype observed in *unc-75* mutant animals (Reviewer 3, point #4), and 4) addressing whether moving the UNC-75 binding site to within 1 nt. of the exon in Fig 5 could disrupt spliceosome assembly independent of UNC-75 binding using an *unc-75* null mutation (Reviewer 1, point #2b and Reviewer 3, point #6). I expect you should be able to submit a revised manuscript within 60 days. A suitably revised manuscript will be acceptable for publication; I don't expect to send it out for review.

Please ensure that you have included a Data Availability Statement at the end of the Materials and Methods section. Details available at <https://academic.oup.com/genetics/content/prep-manuscript>. The DAS should include the accession numbers or DOIs of any data you have placed in public repositories, describe supplemental material, include applicable IRB numbers, and may include specifications for how to properly acknowledge or cite the data.

When revising the manuscript, please make an effort to shorten it, because that almost always improves a manuscript. We urge authors to heed the advice of Strunk and White: "omit needless words". Follow this link to submit the revised manuscript: Link Not Available

Thank you for submitting this story to Genetics.

Sincerely,

Carolyn Phillips
Associate Editor
GENETICS

Approved by:
Karen Arndt
Senior Editor
GENETICS

Reviewer comments:

Reviewer #1 :

Remarks to the authors

In the manuscript titled, "Neuron-specific repression of alternative splicing by the conserved CELF protein UNC-75 in *C. elegans*" the authors report a novel finding that the CELF protein UNC-75 utilizes two critical intronic cis elements-one upstream of exon 9, and the other overlapping the downstream 5' splice site-to facilitate robust exon skipping in neurons. These findings fill important gaps in our understanding of the relationship between the positions of CELF binding sites and splicing regulatory outcomes. Additionally, the study sheds light on how the expression patterns of RNA binding proteins and associated alternative splicing mechanisms contribute to the generation of complex neuron-specific transcriptomic signatures. Moreover, the authors' findings open new opportunities for further exploration in whether this mechanism of tissue specific alternative splicing is true in mammals as well.

Major comments:

1. In Figure 2B, it is good to see that the null mutant derepresses exon 9 in neurons. Does a rescue experiment restore normal function? Employing a conditional mutant such as a temperature sensitive allele of UNC-75 would shed light on the reversibility of this phenotype. The worms would be left at the non-permissive temperature and then switched to permissive temperature then assessed for whether they revert to the normal splicing pattern. The authors do not have to conduct this experiment, but it is good for future consideration.
2. In Figure 5, the authors assess whether the UNC-75 consensus sequence located adjacent to the 5' splice site has a role in exon skipping in neurons in the context of the exon 16 of the *unc-16* locus. It is interesting to observe the position specific effects then the UNC-75 binding site is brought closer to exon 16. However, the experiments that were done do not uncouple the effects of the UNC-75 binding site from those of the EXC-7 site.
 - a. This raises more questions as to what the contribution of the EXC-7 is binding sites to the observed changes in splicing of exon 16. Since EXC-7 RNA binding protein has a role in neuron alternative splicing, it is important for the authors to address this shortcoming and include in the discussion that the observed effects cannot be solely attributed to changes in the UNC-75 binding site alone.
 - b. Additionally, the trimming of the 30nt proximal to the 5' splice site is likely to affect RNA looping that would otherwise facilitate efficient splicing at the locus. Can the authors comment on how the shorted distance especially in the *unc-75* +1 motif could contribute to the observed changes in the alternative splicing of exon 16.
3. The experiments in Figure 5 raises additional questions on whether these distance specific effects would be true at the *zoo-1* UNC-75 locus as well. In Figure S3 when the *zoo-1* exon 9 impacts exon usage was analyzed, the 30nt proximal to the exon 9 splice sites were preserved, if these 30nt were also highly trimmed as in Figure 5, would the effect be the same?
4. The experiments presented address the questions posed by the authors. The proposed model in Figure 6 makes sense. Were co-immunoprecipitation or mass spectrometry experiments done to identify UNC-75 binding partners? Additionally, I would recommend Electrophoretic Mobility Shift Assay and/or Isothermal Titration Calorimetry experiments to further refine the proposed model in Figure 6 in defining whether one or 2 molecules of UNC-75 are binding to the locus.

Minor comments:

1. Page 13 line 10 include citation of the previous work.
2. Page 31, in the figure legend please make it clear that the labels 1, 2, 3A, and 3B correspond to the regions that you reference on page 10 of the manuscript.
3. Does the double region mutant of regions 3A, 3B, and 2 mimic show an additive splicing defect? Rephrasing the question: what is the combined contribution of regions 2 and 3 to exon 9 alternative splicing?
4. In Figure 5, except for the *unc-75* +1 motif, it is difficult to observe the changes in splicing of the Figure 5B gel and I wonder if this was the best locus for this assay.

Reviewer #2 :

This study contains an in-depth analysis of the CELF ortholog UNC-75's role and its key cis-elements that control the splicing patterns of exon 9 in the *C. elegans* *zoo-1* gene in both neuronal and non-neuronal tissues. The author identified conserved intronic UNC-75 motifs near splice sites that regulate a neuron-repressed alternative exon in the gene *zoo-1*. The major findings are that mutations in these motifs or loss of UNC-75 leads to exon de-repression in neurons, and that mis-expression of UNC-75 in muscle can induce neuron-like splicing patterns, demonstrating the significant role of these cis-elements in tissue-specific splicing regulation.

I found this manuscript to be enjoyable to read. It is well-organized, and the figures effectively support the arguments presented. With a few minor revisions, it should be accepted for publication.

- 1) Page 3, line 17: "as species evolve" is nonspecific. Does this change consistently when comparing closely related species, or there are more general alterations to these trans-acting factors when comparing multiple species more broadly?.
- 2) Page 6, line 7: remove the comma between "efficiency" and "and neuronal".
- 3) Page 6, line 7: the last "the" in this line is unnecessary.
- 4) Although color-coded, the minigenes shown in the figures are somewhat confusing. Add text to identify the different isoforms next to the gels.
- 5) Figure 2D: The amount of PCR product recovered in the *unc-75* 2RRM overexpression animals seems to be far less than wild-type, yet it needs to be clarified what method was used to ensure these results are comparable.
- 6) Page 9, line 17: unnecessary comma between "neurons" and "and sufficient".
- 7) Figure 3, region 1: I am unconvinced that this region is "conserved" at all. This issue needs to be clarified in the main text.

- 8) Page 10, paragraph 2: Add more discussion of the cases where conserved sequence elements are present but the homologous protein is not. Why is that? This is an interesting finding and is not discussed in the manuscript.
- 9) Figure 6: To clarify the manuscript's findings, more labeling of relevant exons, introns, and sequence features is needed.

Reviewer #3 :

This manuscript from the Calarco lab explores how UNC-75 (a CELF ortholog) regulates alternative splicing of the zoo-1 exon in neural cells of *C. elegans*. Building on their prior findings, the authors identify dozens of exons with tissue-specific, switch-like regulation. Among these, they focus on zoo-1 exon 9, which is highly included in muscle cells but predominantly skipped in neurons. The authors demonstrate that ectopic expression of the UNC-75 isoform containing three RNA recognition motifs (RRMs) more effectively represses exon 9 inclusion compared to the isoform with two RRM. They also identify conserved UNC-75 binding motifs located upstream and downstream of the regulated exon. Minigene experiments further confirm that these motifs are essential for the regulated skipping of zoo-1 exon 9. Additionally, the authors show that placing an UNC-75 binding motif at the 5' splice site of unc-16 exon 16 is sufficient to repress its inclusion in neurons.

Overall, this study advances our understanding of tissue-specific splicing through the differential expression and activity of RNA-binding proteins. However, addressing the following points would improve the clarity and impact of the work:

1. The authors conclude that the two UNC-75 isoforms exhibit different repressive activities. It is important to compare their expression levels at both RNA and protein levels, as well as their subcellular localization. This would better substantiate the claim of differential repressive activity.
2. The observation that the two UNC-75 isoforms differentially regulate splicing is intriguing. Additional insights into their regulation would enhance the narrative. Are specific neuronal subtypes preferentially expressing or regulating one isoform over the other? This would add valuable context to the findings.
3. It is unclear from the manuscript whether the overexpression experiments were conducted in wild-type animals or in unc-75 null mutants. This distinction should be explicitly stated to avoid confusion.
4. Related to figure 4: The authors assess individual motif disruption; however, it would be informative to also evaluate the effect of disrupting both motifs 2 and 3 simultaneously. Does this recapitulate the splicing phenotype observed in unc-75 null animals?
5. Related to figure S3: The authors raise an interesting point about the importance of intron length in zoo-1 minigene splicing. Testing this hypothesis with randomized intronic sequences could provide additional insights into how sequence context influences splicing regulation.
6. Related to figure 5: It remains unclear whether the observed splicing changes are entirely attributable to UNC-75 activity. Since the authors have access to unc-75 null animals, assessing the behavior of the minigene reporters in this background would clarify the role of UNC-75.
7. Page 17, line 2: If the authors do not plan to reserve certain findings for a future publication, they could consider incorporating them into this manuscript (e.g., as part of Figure 6). This would strengthen the study's conclusions.

Minor comments

1. Page 7, line 16: "and muscle cells" should be probably removed
2. Figure 1B: it would be helpful if the authors indicate the position of the neurons and muscle cells in the brightfield image.
3. Figure 2B: it is unclear what csb20 stands for. This could be moved to the supplementary materials and replaced with "null mutant" in the main text for clarity.
4. The specific mutations introduced to disrupt the UNC-75 binding motifs should be clearly indicated in both the relevant figure legends and the Methods section.
5. Page 13, line 11: The reference citation is missing.

Pilaka-Akela et al., Response to reviewer feedback:

Dear Dr. Calarco:

I am pleased to accept your manuscript entitled "Neuron-specific repression of alternative splicing by the conserved CELF protein UNC-75 in *C. elegans*" for publication in GENETICS, pending minor revision.

Please submit your revision along with a brief description of how you modified the manuscript in response to the reviewers' concerns and suggestions (which can be viewed at the bottom of this email).

Dear Dr. Phillips,

Many thanks to you and the reviewers for the constructive feedback of our study. We were very happy to receive the positive reviews of our manuscript. Below we briefly summarize the most relevant changes suggested by you, and further below provide detailed responses to the points raised by the reviewers.

Most important are:

1) addressing the reason for the lower amount of PCR product recovered in the *unc-75* 2RRM over expression animals and how it was ensured that the results are comparable (Reviewer 2, point #5)

We have tried to clarify why we see differences between different array expressing lines, and describe that we report PSI values, which are internally normalized within each sample as percent isoform values.

2) showing the sub cellular localization and RNA/protein levels of the two UNC-75 isoforms (Reviewer 3, point #1)

We have now added additional data demonstrating that protein expression and subcellular localization of the different UNC-75 isoforms appear to be similar.

3) evaluating whether disrupting both motifs 2 and 3 simultaneously would recapitulate the splicing phenotype observed in *unc-75* mutant animals (Reviewer 3, point #4)

*We have now added additional data demonstrating that disrupting both of these motifs simultaneously leads to an additive effect, more closely recapitulating the phenotype observed in *unc-75* mutants.*

4) addressing whether moving the UNC-75 binding site to within 1 nt. of the exon in Fig 5 could disrupt spliceosome assembly independent of UNC-75 binding using an *unc-75* null mutation (Reviewer 1, point #2b and Reviewer 3, point #6).

*We have now added additional experiments, comparing splicing patterns of the *unc-16* exon 16 reporter containing the UNC-75 5' splice site overlapping cis-element in *unc-75* null mutants. We observe that the exon is still largely skipped in an *unc-75* null background. This suggests in this heterologous context, the splice site strength is significantly weakened, but UNC-75 itself is*

not directly exerting a repressive effect. We have revised our interpretation of these data in the appropriate results and discussion sections.

I expect you should be able to submit a revised manuscript within 60 days. A suitably revised manuscript will be acceptable for publication; I don't expect to send it out for review.

Thank you for your summary of our reviews and for the guidance on how best to complete revisions. We believe we have now addressed these main concerns summarized above and below we provide point by point responses detailing how the manuscript has been revised.

Please ensure that you have included a Data Availability Statement at the end of the Materials and Methods section. Details available at <https://academic.oup.com/genetics/content/prep-manuscript>. The DAS should include the accession numbers or DOIs of any data you have placed in public repositories, describe supplemental material, include applicable IRB numbers, and may include specifications for how to properly acknowledge or cite the data.

We confirm that our Data Availability Statement is up to date

When revising the manuscript, please make an effort to shorten it, because that almost always improves a manuscript. We urge authors to heed the advice of Strunk and White: "omit needless words". Follow this link to submit the revised manuscript: <https://genetics.msubmit.net/cgi-bin/main.plex?el=A5NR3Gfz2A1Kh12I5A9ftdR3fe6B3WVvL6EKg3vnDwZ>

Thank you for submitting this story to Genetics.

Reviewer comments:

Reviewer #1 :

Remarks to the authors

In the manuscript titled, "Neuron-specific repression of alternative splicing by the conserved CELF protein UNC-75 in *C. elegans*" the authors report a novel finding that the CELF protein UNC-75 utilizes two critical intronic cis elements—one upstream of exon 9, and the other overlapping the downstream 5' splice site—to facilitate robust exon skipping in neurons. These findings fill important gaps in our understanding of the relationship between the positions of CELF binding sites and splicing regulatory outcomes. Additionally, the study sheds light on how the expression patterns of RNA binding proteins and associated alternative splicing mechanisms contribute to the generation of complex neuron-specific transcriptomic signatures. Moreover, the authors' findings open new opportunities for further exploration in whether this mechanism of tissue specific alternative splicing is true in mammals as well.

Major comments:

1. In Figure 2B, it is good to see that the null mutant derepresses exon 9 in neurons. Does a rescue experiment restore normal function? Employing a conditional mutant such as a temperature sensitive allele of UNC-75 would shed light on the reversibility of this phenotype. The worms would be left at the non-permissive temperature and then switched to permissive temperature then assessed for whether they revert to the normal splicing pattern. The authors do not have to conduct this experiment, but it is good for future consideration.

This is an interesting future experiment. Although no temperature sensitive alleles exist for unc-75, we could potentially perform the suggested experiment with the auxin-inducible degron system. We will add this to our growing list of upcoming experiments!

2. In Figure 5, the authors assess whether the UNC-75 consensus sequence located adjacent to the 5' splice site has a role in exon skipping in neurons in the context of the exon 16 of the unc-16 locus. It is interesting to observe the position specific effects then the UNC-75 binding site is brought closer to exon 16. However, the experiments that were done do not uncouple the effects of the UNC-75 binding site from those of the EXC-7 site.

a. This raises more questions as to what the contribution of the EXC-7 binding sites to the observed changes in splicing of exon 16. Since EXC-7 RNA binding protein has a role in neuron alternative splicing, it is important for the authors to address this shortcoming and include in the discussion that the observed effects cannot be solely attributed to changes in the UNC-75 binding site alone.

We agree with the reviewer that the perturbation of moving the UNC-75 site further away (more upstream) from the EXC-7 site could also have an impact on regulation due to possible cooperation of UNC-75 and EXC-7 in regulating unc-16 exon 16 splicing. Indeed, in our previous work (Norris et al. Mol. Cell 2014), upon disruption of both binding sites and additional sites in the alternative exon, we dramatically abolished exon inclusion. We have included this shortcoming in the text surrounding this figure.

b. Additionally, the trimming of the 30nt proximal to the 5' splice site is likely to affect RNA looping that would otherwise facilitate efficient splicing at the locus. Can the authors comment on how the shortened distance especially in the unc-75 +1 motif could contribute to the observed changes in the alternative splicing of exon 16.

We apologize for the confusion provided in the design of this experiment. Upon moving the UNC-75 consensus binding motif progressively closer to the 5' splice site, we maintained the overall intron size. Thus, the EXC-7 consensus sequence, remains in its natural position in the intron. We have now clarified the schematic showing the experimental design to indicate the relative positions and distances/sizes of the modified intron sequences. We have also revised the text and results accordingly to better highlight these points.

3. The experiments in Figure 5 raises additional questions on whether these distance specific effects would be true at the zoo-1 UNC-75 locus as well. In Figure S3 when the zoo-1 exon 9

impacts exon usage was analyzed, the 30nt proximal to the exon 9 splice sites where preserved, if these 30nt were also highly trimmed as in Figure 5, would the effect be the same?

As mentioned above, we did not change the overall size of the unc-16 intron in the experiments in Figure 5. However, it would be an interesting experiment to start manipulating both the position of the UNC-75 cis element and the overall length of the intron to determine if these two variables act in additive or synergistic manners. We hope the new text and labeling will help clarify the nature of the experiments presented.

4. The experiments presented address the questions posed by the authors. The proposed model in Figure 6 makes sense. Were co-immunoprecipitation or mass spectrometry experiments done to identify UNC-75 binding partners? Additionally, I would recommend Electrophoretic Mobility Shift Assay and/or Isothermal Titration Calorimetry experiments to further refine the proposed model in Figure 6 in defining whether one or 2 molecules of UNC-75 are binding to the locus.

These suggestions are all very interesting experiments. We hope to perform these in the future to better clarify the mechanism of UNC-75 action at this splicing event, and possibly more broadly.

Minor comments:

1. Page 13 line 10 include citation of the previous work.
we have now included a citation (Norris et al. 2014).

2. Page 31, in the figure legend please make it clear that the labels 1, 2, 3A, and 3B correspond to the regions that you reference on page 10 of the manuscript.
We have now revised the figure legend to better connect the definition of Regions 1, 2, and 3A/3B to the text in the results section.

3. Does the double region mutant of regions 3A, 3B, and 2 mimic show an additive splicing defect? Rephrasing the question: what is the combined contribution of regions 2 and 3 to exon 9 alternative splicing?
This is an interesting question and was also raised by Reviewer #3. We have now generated mutations in all three cis elements, and the results are now presented in Figure S4. Indeed, it appears that disruption of regions 3A, 3B, and 2 simultaneously creates an additive effect, more closely mimicking the splicing pattern found in neurons of unc-75 null animals. We thank the reviewers for suggesting this experiment.

4. In Figure 5, except for the unc-75 +1 motif, it is difficult to observe the changes in splicing of the Figure 5B gel and I wonder if this was the best locus for this assay.
We agree that in the future it might be better to start with another model locus, perhaps with near complete inclusion in neurons, in order to further investigate the sufficiency of weakening exon inclusion by placing UNC-75 binding sites overlapping the 5' splice site.

Reviewer #2 :

This study contains an in-depth analysis of the CELF ortholog UNC-75's role and its key cis-

elements that control the splicing patterns of exon 9 in the *C. elegans* zoo-1 gene in both neuronal and non-neuronal tissues. The author identified conserved intronic UNC-75 motifs near splice sites that regulate a neuron-repressed alternative exon in the gene zoo-1. The major findings are that mutations in these motifs or loss of UNC-75 leads to exon de-repression in neurons, and that mis-expression of UNC-75 in muscle can induce neuron-like splicing patterns, demonstrating the significant role of these cis-elements in tissue-specific splicing regulation. I found this manuscript to be enjoyable to read. It is well-organized, and the figures effectively support the arguments presented. With a few minor revisions, it should be accepted for publication.

1) Page 3, line 17: "as species evolve" is nonspecific. Does this change consistently when comparing closely related species, or there are more general alterations to these trans-acting factors when comparing multiple species more broadly?.

We have tried to clarify this concept—the reviewer is correct in that trans-acting factor alterations in abundance/activity tend to evolve over longer timescales compared to cis element evolution.

2) Page 6, line 7: remove the comma between "efficiency" and "and neuronal".

Comma removed

3) Page 6, line 7: the last "the" in this line is unnecessary.

"The" removed

4) Although color-coded, the minigenes shown in the figures are somewhat confusing. Add text to identify the different isoforms next to the gels.

We apologize for the confusion and we have now added text to label the different isoforms

5) Figure 2D: The amount of PCR product recovered in the unc-75 2RRM overexpression animals seems to be far less than wild-type, yet it needs to be clarified what method was used to ensure these results are comparable.

In the UNC-75 overexpression experiments transgenic F1 animals undergo larval arrest, and we were limited with the material we could collect among F1 animals. Additionally, the amount of reporter amplified is a result of the number of transgenic animals present at the time of collection and the number of cells expressing the reporters within each animal. It is thus difficult to control for overall signal generated in these experiments. However, the PSI value measurements are internally normalized from sample to sample because the measurement calculates the percent exon inclusion within each sample. Thus, PSI values can be directly compared independent of overall signal abundance, provided they can be faithfully detected on the gel and densitometry can be performed. We have tried to clarify these details in the Materials and Methods section of the manuscript, because we feel that these nuances might overly complicate the description of the results.

6) Page 9, line 17: unnecessary comma between "neurons" and "and sufficient".

We have revised this sentence.

7) Figure 3, region 1: I am unconvinced that this region is "conserved" at all. This issue needs to be clarified in the main text.

We agree that this region is at best infrequently containing some matching UNC-75 consensus sequences. We have revised the text to indicate that we took a liberal approach to define regions for subsequent mutagenesis experiments.

8) Page 10, paragraph 2: Add more discussion of the cases where conserved sequence elements are present but the homologous protein is not. Why is that? This is an interesting finding and is not discussed in the manuscript.

In discussions with our collaborator, it is likely that UNC-75 orthologs in these species exist. However, due to poor/incomplete genome sequence information, or homology finding algorithms splitting open reading frames into multiple accessions, this made it difficult to include some of these putative orthologous sequences in the final alignment. We have added details about these nuances in the Materials and Methods section of the manuscript. We have also revised the text in the corresponding results section.

9) Figure 6: To clarify the manuscript's findings, more labeling of relevant exons, introns, and sequence features is needed.

As requested by other reviewers as well, we have attempted to clarify diagrams and schematics in the figures. We hope this improves clarity.

Reviewer #3 :

This manuscript from the Calarco lab explores how UNC-75 (a CELF ortholog) regulates alternative splicing of the zoo-1 exon in neural cells of *C. elegans*. Building on their prior findings, the authors identify dozens of exons with tissue-specific, switch-like regulation. Among these, they focus on zoo-1 exon 9, which is highly included in muscle cells but predominantly skipped in neurons. The authors demonstrate that ectopic expression of the UNC-75 isoform containing three RNA recognition motifs (RRMs) more effectively represses exon 9 inclusion compared to the isoform with two RRM. They also identify conserved UNC-75 binding motifs located upstream and downstream of the regulated exon. Minigene experiments further confirm that these motifs are essential for the regulated skipping of zoo-1 exon 9. Additionally, the authors show that placing an UNC-75 binding motif at the 5' splice site of unc-16 exon 16 is sufficient to repress its inclusion in neurons.

Overall, this study advances our understanding of tissue-specific splicing through the differential expression and activity of RNA-binding proteins. However, addressing the following points would improve the clarity and impact of the work:

1. The authors conclude that the two UNC-75 isoforms exhibit different repressive activities. It is important to compare their expression levels at both RNA and protein levels, as well as their subcellular localization. This would better substantiate the claim of differential repressive activity.

We thank the reviewer for this important suggestion. Given the difficulty in obtaining transgenic animals expressing each isoform due to F1 progeny arresting as larvae, it was difficult to monitor protein levels through Western blotting. Moreover, our original transgenes contained a

2A peptide in between the GFP tag and UNC-75 proteins, preventing us from observing the subcellular localization of the proteins. We thus attempted to address these points by creating new N-terminal GFP::UNC-75 fusion expression constructs. We confirmed by microscopy that the 2RRM and 3RRM-containing protein isoforms are expressed at similar levels in neurons when injected at similar concentrations in over-expression experiments. We also observed by confocal microscopy that both isoforms are nuclear localized, with the previously observed concentration in several dynamic nuclear foci. We have now added this data in Figure S2 and describe these results in the appropriate section.

2. The observation that the two UNC-75 isoforms differentially regulate splicing is intriguing. Additional insights into their regulation would enhance the narrative. Are specific neuronal subtypes preferentially expressing or regulating one isoform over the other? This would add valuable context to the findings.

This is also an important point. Since our previous submission, we have now referenced two new pre-prints analyzing individual neuron subtype transcriptome data. These studies revealed interesting differential alternative splicing patterns among individual classes of neurons. Specifically, these authors highlight varying isoform ratios of unc-75 2RRM and 3RRM encoding transcripts. We now include these interesting results in our discussion section and cite the relevant study.

3. It is unclear from the manuscript whether the overexpression experiments were conducted in wild-type animals or in unc-75 null mutants. This distinction should be explicitly stated to avoid confusion.

We thank the reviewer for this point. We have now clarified that section of the results to more explicitly state that the overexpression experiments are performed in strains that express endogenous UNC-75.

4. Related to figure 4: The authors assess individual motif disruption; however, it would be informative to also evaluate the effect of disrupting both motifs 2 and 3 simultaneously. Does this recapitulate the splicing phenotype observed in unc-75 null animals?

This is an interesting question and was also raised by Reviewer #1. We have now generated mutations in all three cis elements, and the results are now presented in Figure S4. Indeed, it appears that disruption of regions 3A, 3B, and 2 simultaneously creates an additive effect, more closely mimicking the splicing pattern found in neurons of unc-75 null animals. We thank the reviewers for suggesting this experiment.

5. Related to figure S3: The authors raise an interesting point about the importance of intron length in zoo-1 minigene splicing. Testing this hypothesis with randomized intronic sequences could provide additional insights into how sequence context influences splicing regulation.

We agree that this will be an interesting question to pursue in future studies, both in the specific context of this minigene, and more broadly in a genome-wide context. Our previous work studying alternative splicing patterns genome-wide (Koterniak et al. 2020) has suggested that, in general, introns flanking alternative exon, are longer than those flanking constitutively spliced exons.

6. Related to figure 5: It remains unclear whether the observed splicing changes are entirely attributable to UNC-75 activity. Since the authors have access to *unc-75* null animals, assessing the behavior of the minigene reporters in this background would clarify the role of UNC-75.

*We thank the reviewer for suggesting this experiment. We have now included data assessing whether loss of UNC-75 activity impacts repression seen when inserting an UNC-75 consensus sequence at the 5' splice site flanking the *unc-16* alternative exon (Figure 5C). Even in the absence of *unc-75*, we still observe robust skipping of the alternative exon rather than de-repression. These results suggest that at this heterologous locus, the presence of an overlapping *unc-75* cis element can weaken the 5' splice site independently of UNC-75 activity. We have revised the results section to accommodate these new findings. We also place these results in a comparative context with the *zoo-1* locus in the discussion, and we have also revised the subheading title of this section.*

7. Page 17, line 2: If the authors do not plan to reserve certain findings for a future publication, they could consider incorporating them into this manuscript (e.g., as part of Figure 6). This would strengthen the study's conclusions.

We agree with the reviewer that this is an important result. As the reviewer suggests, we are planning on expanding this result further through more precise spatio-temporal mis-expression of UNC-75 to assess its ability to alter developmental programs and the lethality we are observing with multi-copy array-based overexpression.

Minor comments

1. Page 7, line 16: "and muscle cells" should be probably removed

We have removed "and muscle cells"

2. Figure 1B: it would be helpful if the authors indicate the position of the neurons and muscle cells in the brightfield image.

We have now highlighted the positions of some of the neurons and muscle cells on the brightfield and single fluorescent channel micrographs to give perspective on locations.

3. Figure 2B: it is unclear what *csb20* stands for. This could be moved to the supplementary materials and replaced with "null mutant" in the main text for clarity.

*We have changed *csb20*, which is an allele designator, to "null" in Figure 2B.*

4. The specific mutations introduced to disrupt the UNC-75 binding motifs should be clearly indicated in both the relevant figure legends and the Methods section.

We have now included better labelling of the mutations in Figure 3 and its accompanying legend. In the Figure 4 legend, we also reference the diagram and notations in Figure 3 to provide better clarity on the sequences mutated. We have also included additional text in the Methods section to provide the specific sequence regions and precise nucleotides mutated, again making reference to Figure 3 as an accompanying diagram.

5. Page 13, line 11: The reference citation is missing.

We have now included the missing citation.

January 29, 2025

RE: GENETICS-2024-307490R1

Prof. John A. Calarco
University of Toronto
Cell and Systems Biology
25 Harbord Street
Room 513
Toronto, N/A M5S 3G5
Canada

Dear Dr. Calarco:

Congratulations! We are delighted to inform you that your manuscript entitled "Neuron-specific repression of alternative splicing by the conserved CELF protein UNC-75 in *C. elegans*" is acceptable for publication in GENETICS. Many thanks for submitting your research to the journal.

To Proceed to Production:

1. Format your article according to GENETICS style, as discussed at <https://academic.oup.com/genetics/pages/general-instructions>, and upload your final files at <https://genetics.msubmit.net>.
2. Your manuscript will be published as-is (unedited-as submitted, reviewed, and accepted) at the GENETICS website as an Advanced Access article and deposited into PubMed shortly after receipt of source files and the completed license to publish. Please notify sourcefiles@thegsajournals.org if you do not wish to publish your article via Advanced Access.
3. We invite you to submit an original color figure related to your paper for consideration as cover art. Please email your submission to the editorial office or upload it with your final files. You can submit a small-sized image for evaluation, and if selected, the final image must be a TIFF file 2513px wide by 3263px high (8.375 by 10.875 inches; resolution of 600ppi). Please avoid graphs and small type.

If you have any questions or encounter any problems while uploading your accepted manuscript files, please email the editorial office at sourcefiles@thegsajournals.org.

Sincerely,

Carolyn Phillips
Associate Editor
GENETICS

Approved by:
Karen Arndt
Senior Editor
GENETICS

note: Please add jnls.author.support@oup.com and genetics.oup@kwglobal.com (or the domains @oup.com and @kwglobal.com) to your email program's "safe senders" list. You will be contacted by both at various points during the production process.